# Changes in Woody Vegetation over 31 Years in Farmed Parkland of the Central Plateau, Burkina Faso

**Koichi Takenaka** [1,*], **Kenta Ikazaki** [2], **Saïdou Simporé** [3], **François Kaboré** [3], **Natacha Thiombiano** [3] **and Jonas Koala** [3]

1    Rural Development Division, Japan International Research Center for Agricultural Sciences, 1-1 Ohwashi, Tsukuba, Ibaraki 305-8686, Japan

2    Crop, Livestock and Environment Division, Japan International Research Center for Agricultural Sciences, 1-1 Ohwashi, Tsukuba, Ibaraki 305-8686, Japan; ikazaki@affrc.go.jp

3    Station Agricole de Saria, Institut de l'Environnement et Recherches Agricoles, B.P. 10, Saria, Koudougou 01 BP 476, Burkina Faso; simpsaid@hotmail.fr (S.S.); frankiskabore77@gmail.com (F.K.); thionat@yahoo.fr (N.T.); koalajonas@gmail.com (J.K.)

*    Correspondence: koichitk@affrc.go.jp; Tel.: +81-29-838-6681

**Abstract:** Farmed parklands of the Central Plateau, Burkina Faso, integrate native woody vegetation with managed cropland. However, sapling survival in the parklands is increasingly threatened. This study characterized woody vegetation abundance along a 2.7 km long transect in the Doulou Basin, Boulkiemdé Province, Central West Region, to assess changes in vegetation composition since 1984. In addition, a householder survey was conducted to gain insight into tree uses and preferences and residents' knowledge of regulations. In total, 4999 individuals from 26 tree species were recorded, including 123 individuals (11 species) with stem DBH $\geq$ 5 cm, and 4876 individuals (21 species) with stem DBH < 5 cm. The three species with the highest importance value index provided fruit for sale or self-consumption. Tree abundance was associated with soil type and topography; highest abundance was on Lixisol soils along the lower transect. Soil degradation and preference changes among residents since 1984 may have influenced tree abundance. Certain beneficial species (e.g., *Vitellaria paradoxa*) have declined in abundance, and certain exotics (*Azadirachta indica* and *Eucalyptus camaldulensis*) have expanded in distribution. Respondents expressed strongest interest in three species, including *V. paradoxa*, that show high versatility. These results supported the recorded tree composition. The respondents generally understood forest conservation regulations. Dissemination of regreening technology and awareness promotion among residents is essential for sustainable tree use in farmed parklands.

**Keywords:** degradation; desertification; line transect; native trees; Plinthosols

## 1. Introduction

The composition of woody vegetation can indicate the current tree status and other vegetation characteristics of a landscape. Different species of trees and shrubs may establish plant communities that show considerable variation in age and size [1]. The notion of "parkland" has been widely used, but its definition remains controversial. Cole [2] first defined "savanna parklands", but a variety of additional terms have since been coined by phytogeographers, including "savanna parkland", "park savanna", and "parklike savanna" [3]. Human usage also affects such vegetation communities. For example, "farmed parkland" may refer to cultivated or fallow lands with scattered mature trees that have been conserved in residents' daily life [1]. In many cases, such lands are continuously managed together with staple crops planted among the more sparsely distributed tall trees, but sometimes fallowing is favorable for sapling or coppice regrowth of shrubs [1]. In the present study, we use the term "farmed parkland" to refer to a cultivated area containing sparse trees.

The composition of existing tree communities in farmed parkland can be dramatically altered by human activities and preferences. Pullan [1] observed that "these trees may be attributed directly to dispersion and protection by man", and Tomomatsu [4] concluded that farmed parkland is created through deliberate manipulation of trees. In addition, the farmed parkland landscape and its preservation are closely associated with the livelihood of rural residents because the dominant tree species of such areas often varies with the resident ethnic group [5,6].

Deforestation and soil degradation caused by rapid population increase and agricultural expansion also affect the composition of woody plants [7]. On the Central Plateau of Burkina Faso, where water erosion is severe [8], soil erosion and soil degradation are important factors that affect the environment, food security, and human livelihoods. Previous studies have reported that crop yields, woody biomass, and natural resources have declined or have been degraded by water and wind erosion [9–11].

The effects of soil erosion may be amplified by the dominant soils in this region, defined as Plinthosols in the current World Reference Base (WRB) soil classification system [12]. According to the *Soil Atlas of Africa* [13], these Plinthosols have a petroplinthic horizon (an iron hardpan, formerly termed laterite) or pisoplinthic horizon (a layer containing abundant iron nodules) starting ≤50 cm below the surface. Given that these horizons reduce the soil volume available for root elongation and storage of water and nutrients, the crop yield and woody biomass can be severely limited [14,15]. If these horizons are exposed by erosion, soil rehabilitation is difficult. The present-day woody plant communities were primarily established by incidental adaptation to previously prevailing conditions [16] and have been strongly influenced by the present soil conditions and local human land use [7].

In recent years, a close link between climate change and complex vegetation changes has been revealed. Hänke et al. [17] found that the West African Sahel has become greener again after severe droughts in the 1970s–1980s. However, the species composition has substantially changed towards a higher dominance of drought-resistant and exotic species, and there was some debate as to whether increased annual rainfall was the sole primary driver of the increase in tree cover. Zida et al. [18] also found that the post-drought flora of the Sahel region was highly resilient during the end of the 1970s–1980s. In contrast, the diversity and density of woody species had declined, and more drought-resistant woody species were dominant at that time. Furthermore, Brandt et al. [19] pointed out that vegetation cover and plant diversity substantially fluctuate around drought, which affects regional resilience, and woody cover in the Sahel region responds to its inherent climatic variability. All of those studies found that fluctuations in vegetation cover occur at a decadal time scale, and that changes in flora should be observed over a longer period with multiple factors.

Although there are many findings in terms of vegetation changes in parkland and the factors affecting it in Sahelian countries, we considered that it would be meaningful to share the results of actual soil sampling, tree surveys, and residents' interviews that were collected in this survey. Moreover, a comparison of the results of this study with those of earlier studies, including soil and vegetation classification information gathered in the same area of this study, can provide valuable regional environmental information. In this study, we first assessed woody vegetation communities in a farmed parkland, and then interpreted the effects of human activities and soil conditions during the last 31 years by comparing our results with those of previous studies [20,21].

## 2. Materials and Methods

### 2.1. Research Site

The study was conducted in August 2015, encompassing a farmed parkland near the village of Villy in the Doulou Basin, Boulkiemdé Province, Central West Region, Burkina Faso (Figures 1 and 2). The basin's mean altitude is approximately 300 m above sea level (m a.s.l.) with mostly level land (mean slope 1%). Owing to the semi-arid climate, the main crops grown are sorghum (*Sorghum bicolor*), pearl millet (*Pennisetum glaucum*), cowpea

(*Vigna unguiculata*), and groundnut (*Arachis hypogaea*); rice (*Oryza* spp.) and vegetables are only grown in the valley bottom under rainfed systems. The farmed parkland in this area is typical of Sudanian savanna as well as the Sahel region. This landscape is classified as "other wooded land" (sparse woody vegetation covering 17.5% of the national territory) or "other land (with tree cover)" (22.3% of the land area) [22].

The region's average maximum and minimum temperatures are 32.1 and 25.0 °C, respectively, with a mean annual temperature of 28.1 °C. Dry and wet seasons are clearly distinguishable with approximately 80% of annual precipitation occurring from June to September; the mean annual rainfall is 782 mm year$^{-1}$ (all meteorological data for 1982–2012) [23].

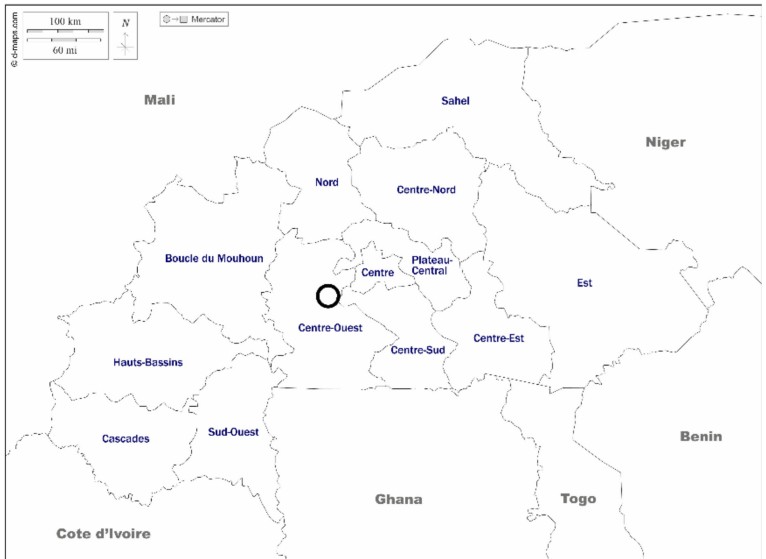

**Figure 1.** Study site location (circle) in the village of Villy, Central West region, Republic of Burkina Faso [24].

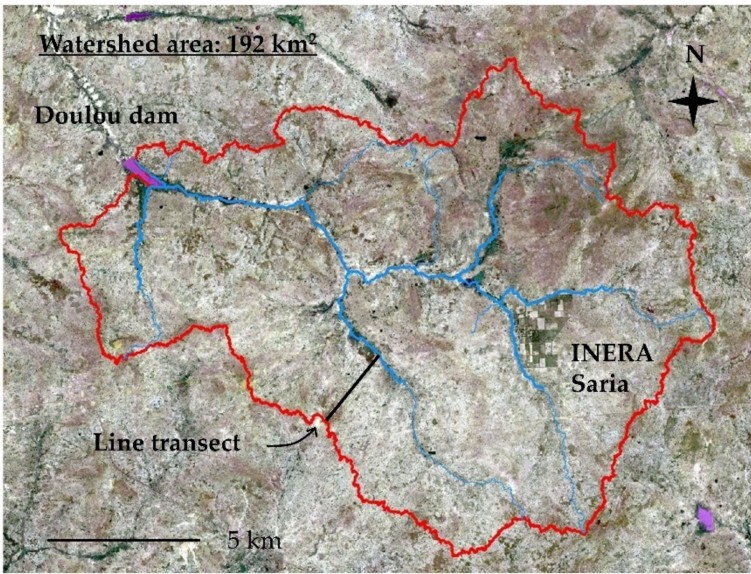

**Figure 2.** Line transect location in the Doulou Basin with watershed boundary (red line), seasonally dry riverbed locations (blue lines), and water bodies of dams or reservoirs (purple polygons). Location of the Institut de l'Environnemont et Recherches Agricoles (INERA) Saria research station is indicated. Image source: RapidEye AG, 2013.

### 2.2. Soil Survey

We first established a 2.7 km long line transect from the plateau to the valley bottom (Figure 2). Topography data were collected using two Global Navigation Satellite System receivers (ProMark 100, Spectra Precision, Westminster, CO, USA) as a base and a rover to obtain raw data at 1 s intervals. Post-processing data analysis was conducted using GNSS Solutions version 3.8 software (Trimble Navigation, Sunnyvale, CA, USA). The mean vertical error (two-sided 95% confidence interval) was <0.08 m. Along the transect, we placed 54 adjacent quadrats of 50 × 50 m (13.5 ha in total) (Figure 3). We could not set a quadrat at the center of the valley bottom because of waterlogging.

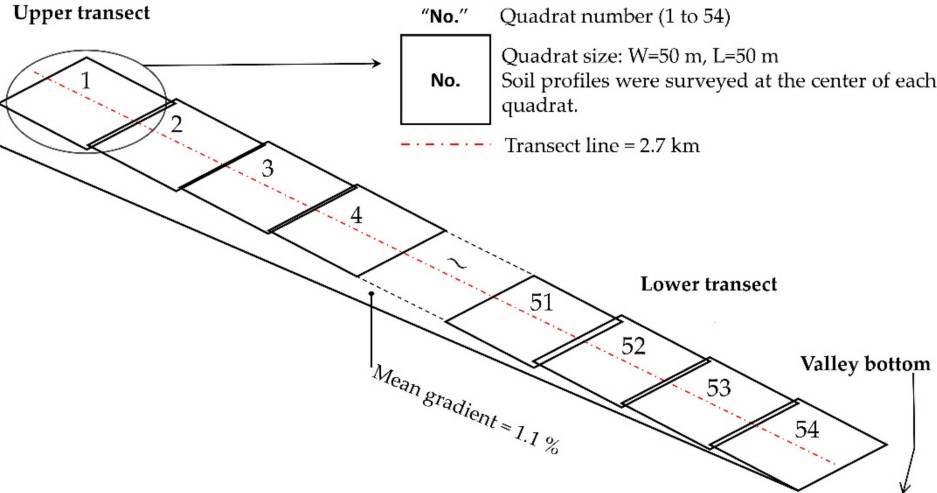

**Figure 3.** Schematic illustration of the line transect.

We examined soil profiles at the center of each quadrat using a hand auger (One-piece Combination Auger, Eijkelkamp, Giesbeek, The Netherlands) and roughly classified the soil type using the current WRB system [12]. In each soil class, a soil pit was dug to describe the profile following the *Guidelines for Soil Description* [25] and to obtain soil samples for physical and chemical analyses. We measured the particle size distribution, bulk density, pH ($H_2O$ and 1 M KCl), electrical conductivity (EC), organic carbon (OC), total nitrogen (TN), exchangeable bases, and available phosphorus (Bray-I method). We then confirmed the auger-based soil classifications using the soil profile descriptions and soil physical and chemical properties. Effective soil depth was also measured in each quadrat, defined as the soil depth overlying a petroplinthic or pisoplinthic horizon.

### 2.3. Woody Vegetation Survey

Within each quadrat, the woody plants were counted and preliminarily identified by a researcher from the Institute of the Environment and Agricultural Research (INERA), and subsequently confirmed using an online database [26] and a tree species guide [27]. The stem diameter at breast height (1.3 m, DBH; cm), tree height (*H*; m), and canopy width in two orthogonally crossed lines ($CW_1$ and $CW_s$; cm) were measured for trees of DBH ≥ 5 cm. For trees of DBH < 5 cm, only $CW_1$ and $CW_s$ were measured for 52 randomly selected woody plants to estimate canopy coverage as a reference. Mean canopy coverage was calculated from the data for the 52 samples of shrubs of DBH < 5 cm from three species comprising *Combretum* spp., *Guiera senegalensis*, and *Piliostigma reticulatum*. The mean canopy coverage was calculated for the total of 4876 individuals that were identified in the transect. Canopy coverage per hectare for trees of DBH < 5 cm was estimated from the mean canopy coverage for the 4876 individuals and the total study area (13.5 ha). Data for DBH, $CW_1$ and $CW_s$, and *H* were gathered using a diameter tape (DM-5, Taketani Trading, Osaka, Japan), a 5.5 m measuring tape (off-brand: of first accuracy quality in accordance

with the Japanese Industrial Standards), and a height measurement instrument (Vertex III, Haglöf, Långsele, Sweden), respectively.

### 2.4. Vegetation Parameters

We used the vegetation survey data to determine the basal area at breast height (BA; $m^2$), canopy coverage (CC; $m^2$), importance value index (IVI) [28] for trees of DBH $\geq$ 5 cm (because other woody plants of DBH < 5 cm can be less than 1.3 m in height), Shannon diversity index ($H'$), and Pielou evenness index ($E$) for all trees and shrubs, using the following formulae:

$$BA = \pi[(DBH/2)]^2, \tag{1}$$

where DBH was measured at 1.3 m height above the ground,

$$CC = \pi \times (CW_l/2) \times (CW_s/2), \tag{2}$$

where $CW_l$ and $CW_s$ are two orthogonally crossed lines within the canopy,

$$IVI = RD + RF + Rdom, \tag{3}$$

where RD, RF, and Rdom are relative density, relative frequency, and relative dominance, respectively, as defined by the following formulae:

$$RD\ (\%) = (\text{Density of a certain species})/(\text{Density of all species}) \times 100, \tag{4}$$

where density is the ratio of number of individuals to survey area,

$$RF\ (\%) = (\text{Frequency of a certain species})/(\text{Total frequency of all species}) \times 100, \tag{5}$$

where frequency is the ratio of number of quadrats containing a species to total number of quadrats,

$$Rdom\ (\%) = (\text{Dominance of a certain species})/(\text{Dominance of all species}) \times 100, \tag{6}$$

where dominance is the ratio of BA of a species to survey area,

$$H' = \sum_{(i=1)}^{R} (pi \times \ln\ pi), \tag{7}$$

where *pi* is the proportion of the *i*th individual species in relation to all species and *R* is the total number of species, and

$$E = H'/(\ln S), \tag{8}$$

where *S* is the number of species.

### 2.5. Survey of Residents on Preferred Useful Trees

We interviewed rural residents on the preferred trees that were used daily in the survey area. With regard to the trees identified along the line transect, it was considered that the distribution was influenced by social conditions, such as the use, preference, and rights to the trees. Following the 2015 tree inventory, semi-structured interviews were conducted in August 2016, targeting 30 householders living within 3 km of the transect. In addition to requesting household information, we asked the tree species considered useful and their uses; local rules and rights regarding tree uses; and conflict between residents on tree uses.

In Question 1, we asked each respondent what kinds of tree they prefer to use and summarized the responses using the scientific name of the species. The number of responses was limited to a maximum of 10 to rank the importance of preferences for each respondent. We presented options in advance for uses of each tree, which were categorized as (1) for

construction, (2) for commodities, (3) for fuel, (4) edible, (5) livestock feed, (6) traditional medicine, and (7) other uses, and multiple answers were allowed. From these results, the total number of multiple uses specified by the respondents for one tree species was divided by the total number of respondents who specified uses of the trees, which was termed the "versatility" for descriptive purposes. Those trees with a high versatility index were indicated to be frequently used or highly valued by the residents and to have many uses.

In Question 2, we queried the residents' perceptions and knowledge of national forest laws, ordinances, and local customary law. We also asked about the respondents' perceptions of the types of actions that are permitted and prohibited in connection to trees in the farmed parkland of the village under the legal regulations.

In Question 3, we asked whether there had been any conflict between stakeholders, such as among residents, on the uses of trees in the farmed parkland. Information on when, who was involved, and the cause of the conflict was also gathered.

### 2.6. Comparison with Previous Research

In this study, a tree inventory survey was planned at the line transect. These data only represent the situation at the time of the survey in 2015 and were considered useful for comparison with previous studies. In 1984, Guinko [20] published a phytogeographic classification based on plant sampling in each region of Haute-Volta (present-day Burkina Faso). Moreover, in 1995, Zerbo [21] evaluated the soil characteristics in the area covered by the INERA-Saria (including the village of Villy, the site of this study). The geographical and topographical conditions have been summarized and contrasted with the vegetation classification reported by Guinko [20].

Since the topographical conditions were based on the positional relationship on the slope and the range of the soil layer, we used our results compared with the previous tree classification reported by Guinko [20] for verifying the presence and/or absence (i.e., "changes") of the tree species between two moments. By comparing the results with the table, they were expressed as 31-year changes in tree species in a specific area. Based on the above, we tried to correlate the vegetation situation in 1984 with the data obtained in our survey in 2015 as a 31-year change.

## 3. Results

### 3.1. Composition of the Woody Vegetation

A total of 4999 individual woody plants (trees or shrubs) and 26 species were identified, of which approximately 95% were native to Sub-Saharan Africa (Table 1). Of these individuals, 123 were trees of DBH $\geq$ 5 cm (11 species), and the remainder were small trees or shrubs (21 species). The remaining 4876 were small trees and shrubs with stem DBH < 5 cm (21 species). Six species overlapped between the two stem DBH classifications. The overall density and CC of trees/shrubs was 370.3 individuals ha$^{-1}$ and 687.3 m$^2$ ha$^{-1}$ (6.9%), respectively; comparable values for trees of DBH $\geq$ 5 cm were 9.1 individuals ha$^{-1}$ and 421.2 m$^2$ ha$^{-1}$ (4.2%). The total BA of trees of DBH $\geq$ 5 cm was 1.6 m$^2$ ha$^{-1}$.

Since 123 individuals (11 spp.) had DBH $\geq$ 5 cm, the IVI for 123 individuals was calculated for each species. The IVI varied from 3.1 to 114.3 (Table 2). The three top-ranked species in terms of IVI were *Vitellaria paradoxa* (114.3), *Lannea microcarpa* (66.5), and *Parkia biglobosa* (54.2), all of which are common native trees that provide fruit for sale and self-consumption by rural residents. The dominant shrub species in terms of number of individual plants were *Guiera senegalensis* (68.5% of total), *Piliostigma reticulatum* (8.8% of total), and *Combretum micranthum* (4.3% of total) (Table 1).

**Table 1.** Composition of woody vegetation along the 54-quadrat transect (total area = 13.5 ha).

| Taxon | Family | Plant Form | Total No. of Individuals | | |
|---|---|---|---|---|---|
| | | | DBH $\geq$ 5 cm | DBH < 5 cm | Total |
| *Azadirachta indica* | MELIACEAE | Tree | 8 | 84 | 92 |
| *Cochlospermum* sp. | COCHLOSPERMACEAE | Tree/Shrub | 0 | 4 | 4 |
| *Combretum glutinosum* | COMBRETACEAE | Tree/Shrub | 0 | 8 | 8 |
| *Combretum micranthum* | COMBRETACEAE | Tree/Shrub | 0 | 217 | 217 |
| *Daniellia oliveri* | LEGUMINOSAE | Tree | 0 | 34 | 34 |
| *Detarium microcarpum* | LEGUMINOSAE | Tree | 0 | 3 | 3 |
| *Diospyros mespiliformis* | EBENACEAE | Tree | 3 | 169 | 172 |
| *Eucalyptus camaldulensis* | MYRTACEAE | Tree | 0 | 182 | 182 |
| *Feretia apodanthera* | RUBIACEAE | Shrub | 0 | 26 | 26 |
| *Ficus* sp. | MORACEAE | Tree | 0 | 1 | 1 |
| *Guiera senegalensis* | COMBRETACEAE | Shrub | 0 | 3426 | 3426 |
| *Khaya senegalensis* | MELIACEAE | Tree | 2 | 0 | 2 |
| *Lannea microcarpa* | ANACARDIACEAE | Tree | 36 | 72 | 108 |
| *Maytenus senegalensis* | CELASTRACEAE | Tree/Shrub | 0 | 1 | 1 |
| *Parkia biglobosa* | LEGUMINOSAE | Tree | 14 | 0 | 14 |
| *Piliostigma reticulatum* | LEGUMINOSAE | Tree/Shrub | 6 | 433 | 439 |
| *Saba senegalensis* | APOCYNACEAE | Shrub/Liana | 0 | 2 | 2 |
| *Sclerocarya birrea* | ANACARDIACEAE | Tree | 2 | 3 | 5 |
| *Senegalia pennata* | LEGUMINOSAE | Shrub | 0 | 46 | 46 |
| *Sterculia* sp. | STERCULIACEAE | Tree | 2 | 0 | 2 |
| *Tamarindus indica* | LEGUMINOSAE | Tree | 1 | 0 | 1 |
| *Terminalia* sp. | COMBRETACEAE | Tree | 2 | 87 | 89 |
| *Vachellia seyal* | LEGUMINOSAE | Tree | 0 | 24 | 24 |
| *Vitellaria paradoxa* | SAPOTACEAE | Tree | 47 | 0 | 47 |
| *Waltheria indica* | STERCULIACEAE | Shrub | 0 | 1 | 1 |
| *Ximenia americana* | OLACACEAE | Tree/Shrub | 0 | 53 | 53 |
| Total (26 spp.) | | | 123 | 4876 | 4999 |
| Number of species (spp.) [†] | | | 11 | 21 | 26 (6) |
| Density of woody plants (individuals ha$^{-1}$) [‡] | | | 9.1 | 361.2 | 370.3 |
| Basal area at breast height (m$^2$ ha$^{-1}$) | | | 1.6 | N/A | N/A |
| Canopy coverage (m$^2$ ha$^{-1}$) | | | 421.2 | 266.1 | 687.3 |
| Shannon diversity index ($H'$) | | 1.36 | - | - | - |
| Pielou's evenness ($E$) | | 0.41 | - | - | - |

[†] Value in parenthesis indicates number of species overlapped in both *DBH* classifications. [‡] Density of woody plants with DBH < 5 cm was estimated from 20 data samples.

**Table 2.** Growth parameters and importance value index of existing woody plants with diameter at breast height $\geq$ 5 cm.

| Taxon | Total No. of Individuals | DBH (cm) [*1, †] | Height (m) [†] | Canopy Coverage [†] (m$^2$ Individual$^{-1}$) | Canopy Coverage (m$^2$ ha$^{-1}$) | Basal Area (m$^2$ ha$^{-1}$) | RD [*2] | RF [*3] | *Rdom* [*4] | IVI [*5] |
|---|---|---|---|---|---|---|---|---|---|---|
| *Azadirachta indica* | 8 | 24.7 ± 3.1 | 6.3 ± 0.8 | 19.8 ± 6.1 | 11.7 | 0.03 | 6.5 | 9.2 | 2.0 | 17.7 |
| *Diospyros mespiliformis* | 3 | 22.5 ± 9.5 | 4.4 ± 0.6 | 0.3 ± 0.0 | 0.1 | 0.01 | 2.4 | 3.1 | 0.7 | 6.3 |
| *Khaya senegalensis* | 2 | 66.0 ± 15.0 | 8.0 ± 0.8 | 15.3 ± 2.8 | 2.3 | 0.05 | 1.6 | 1.5 | 3.3 | 6.5 |
| *Lannea microcarpa* | 36 | 25.2 ± 3.1 | 6.1 ± 0.4 | 23.1 ± 4.2 | 61.5 | 0.20 | 29.3 | 24.6 | 12.7 | 66.5 |
| *Parkia biglobosa* | 14 | 70.3 ± 6.2 | 12.2 ± 0.7 | 141.0 ± 17.6 | 146.2 | 0.44 | 11.4 | 15.4 | 27.4 | 54.2 |
| *Piliostigma reticulatum* | 6 | 43.3 ± 14.3 | 7.7 ± 1.7 | 79.9 ± 33.7 | 35.5 | 0.10 | 4.9 | 4.6 | 6.2 | 15.7 |
| *Sclerocarya birrea* | 2 | 38.3 ± 7.3 | 5.6 ± 0.3 | 31.3 ± 8.7 | 4.6 | 0.02 | 1.6 | 3.1 | 1.1 | 5.8 |
| *Sterculia* sp. | 2 | 49.5 ± 12.5 | 7.1 ± 1.0 | 32.9 ± 15.8 | 4.9 | 0.03 | 1.6 | 3.1 | 1.9 | 6.6 |
| *Tamarindus indica* | 1 | 45.0 | 9.5 | 60.3 | 4.5 | 0.01 | 0.8 | 1.5 | 0.7 | 3.1 |
| *Terminalia* sp. | 2 | 16.8 ± 0.3 | 3.8 ± 0.6 | 6.0 ± 0.4 | 0.9 | < 0.01 | 1.6 | 1.5 | 0.2 | 3.4 |
| *Vitellaria paradoxa* | 47 | 48.2 ± 2.4 | 8.7 ± 0.4 | 42.8 ± 3.5 | 149.1 | 0.71 | 38.2 | 32.3 | 43.8 | 114.3 |
| Total (11 spp.) | 123 | 41.2 ± 2.2 | 7.9 ± 0.3 | 46.2 ± 4.6 | 421.2 | 1.60 | 100.0 | 100.0 | 100.0 | 300.0 |

[†] Values are mean ± standard error. [*1] Diameter at breast height (1.3 m above the ground); [*2] relative density; [*3] relative frequency; [*4] relative dominance; [*5] importance value index, sum of RD, RF, and Rdom.

### 3.2. Soil Distribution and Properties

A distinct soil toposequence was observed along the transect (Figure 4). In general, Pisoplinthic petric Plinthosols (PT-pt.px) composed of both petroplinthic and pisoplinthic horizons were observed along the upper transect. Petric Plinthosols (PT-pt) with a petro-plinthic horizon were present along the middle transect. Pisoplinthic Plinthosols (PT-px) with a pisoplinthic horizon were found along the lower transect, and Lixisols (LX) and Gleysols (GL) that lacked petroplinthic or pisoplinthic horizons were present along the lower transect and valley bottom, respectively. The overall coverage of each soil type was 24% (PT-pt.px), 39% (PT-pt), 19% (PT-px), 15% (LX), and 4% (GL). Overall, 81% of soils in the study area were Plinthosols with low effective soil depth. Soil physical and chemical properties of selected soil profiles from the five soil types are summarized in Table 3. All soils were characterized by a sandy topsoil with low carbon and nutrient contents. In contrast, the effective soil depth was highly variable among soils (Table 3).

**Table 3.** Relationships between topographic position, soil type, woody vegetation, and changes in dominant species in the last 31 years.

| Position | Soil Type | Effective Soil Depth [1], [2] | Mean Number of Individuals [2] | Mean Number of Species [2] | Dominant Species in 1984 | Dominant Species in 2015 [3] |
|---|---|---|---|---|---|---|
| | | (cm) | (Plants Quadrat$^{-1}$) | (Number Quadrat$^{-1}$) | [20,21] [4] | (This Study) |
| Valley bottom | Gleysols (2) | >100 | 175.0 ± 121.0 $^{ab}$ | 8.0 ± 2.0 $^{ns}$ | Anogeissus leiocarpus Butyrospermum paradoxum subsp. parkii [5] Sclerocarya birrea Lannea microcarpa Diospyros mespiliformis | Diospyros mespiliformis Guiera senegalensis Piliostigma reticulatum Vachellia seyal Azadirachta indica |
| Lower transect | Lixisols (8) | >100 | 390.5 ± 172.7 $^{b}$ | 5.8 ± 0.6 $^{ns}$ | | Guiera senegalensis Piliostigma reticulatum Combretum micranthum Terminalia sp. Ximenia americana |
| | Pisolithic Plinthosols (10) | 16.5 ± 4.2 | 20.7 ± 6.5 $^{a}$ | 3.9 ± 0.7 $^{ns}$ | | Guiera senegalensis Piliostigma reticulatum Lannea microcarpa Azadirachta indica Combretum micranthum |
| Middle transect | Petric Plinthosols (21) | 45.3 ± 3.7 | 45.6 ± 13.2 $^{a}$ | 4.6 ± 0.4 $^{ns}$ | Butyrospermum paradoxum subsp. parkii [5] Guiera senegalensis Gardenia erubescens Senegalia macrostachya Piliostigma reticulatum Combretum glutinosum Adansonia digitata Lannea microcarpa Combretum micranthum Ximenia americana | Guiera senegalensis Eucalyptus camaldulensis Piliostigma reticulatum Diospyros mespiliformis Lannea microcarpa |
| Upper transect | Pisolithic Petric Plinthosols (13) | 4.3 ± 2.4 | 27.8 ± 8.4 $^{a}$ | 3.8 ± 0.8 $^{ns}$ | Senegalia macrostachya Guiera senegalensis Piliostigma reticulatum Gardenia erubescens | Guiera senegalensis Piliostigma reticulatum Combretum micranthum Senegalia pennata Lannea microcarpa |

[1] Effective soil depth is defined as the depth of soil overlying a petroplinthic or pisoplinthic horizon. [2] Values are mean ± standard error; different superscript letters indicate a significant difference using the Tukey–Kramer test ($\alpha$ = 0.05); "ns" indicates non-significant. [3] Five tree/shrub species listed in each topographic position/soil type are from all surveyed individuals including DBH ≥ 5 cm and DBH < 5 cm. [4] Vegetation survey data in [20] were cited by [21]. [5] Butyrospermum paradoxum subsp. parkii (G.Don) Hepper is a synonym of Vitellaria paradoxa C.F.Gaertn.

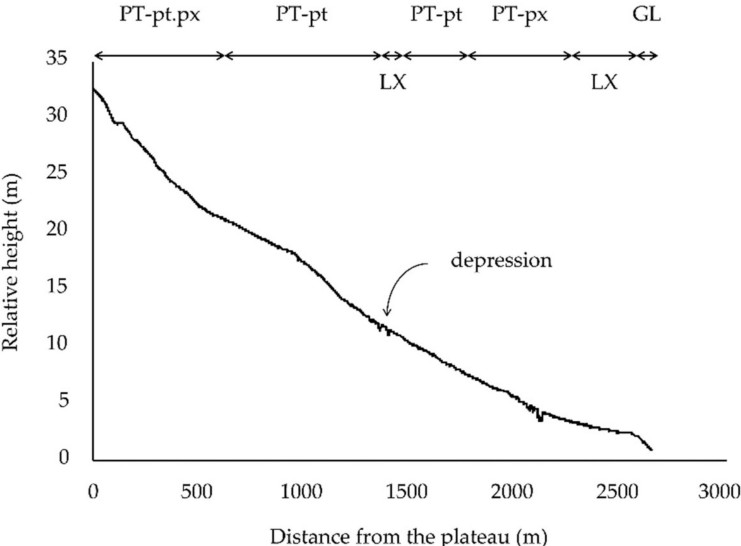

**Figure 4.** Soil toposequence along the line transect. PT−pt.px: pisoplinthic petric Plinthosols; PT−pt: petric Plinthosols; PT−px: pisoplinthic Plinthosols; LX: Lixisols; and GL: Gleysols.

*3.3. Relationships between Topographic Position, Soil Type, and Woody Vegetation*

Consistent with Zerbo [21], we observed clear relationships between topographic position, soil type, and woody vegetation (Table 3). The mean number of all woody plants (individuals) per quadrat on lower-slope Lixisols was significantly higher than that on Plinthosols (Tukey–Kramer test, $p < 0.01$). The mean number of individuals of *G. senegalensis*, *P. reticulatum*, *C. micranthum*, *Terminalia* sp., *Ximenia americana*, *Senegalia pennata*, and *Feretia apodanthera* on Lixisols was significantly higher than that on the other four soil types (Table 4). Although the mean number of all woody plants on Gleysols in the valley bottom was not significantly different from that on Lixisols and Plinthosols, the mean number of individuals of *Vachellia seyal*, *Diospyros mespiliformis*, and *P. biglobosa* was significantly higher than that on Lixisols and Plinthosols (Table 4). In contrast, no significant difference in the mean number of species per quadrat was observed among soil types (Table 3).

*Guiera senegalensis* and *P. reticulatum* were dominant in all soil types and all topographic positions (Table 3), and *C. micranthum* was dominant on many soil types and in many topographic positions. Although ~95% of the 4999 individuals were native, two exotic tree species (*Azadirachta indica* and *Eucalyptus camaldulensis*) were dominant on certain soil types and topographic positions.

**Table 4.** Statistical comparison of the mean number of individuals of existing woody plants.

| Taxon | Total Abundance of Woody Plants | Petric Pisolithic Plinthosols N = 13 | Petric Plinthosol N = 21 | Lixisol N = 8 | Pisolithic Plinthosol N = 10 | Gleysols N = 2 |
|---|---|---|---|---|---|---|
| *Azadirachta indica* | 92 | 0.15 a | 0.90 a | 4.38 a | 2.00 a | 8.00 a |
| *Cochlospermum* sp. | 4 | 0.00 a | 0.19 a | 0.00 a | 0.00 a | 0.00 a |
| *Combretum glutinosum* | 8 | 0.54 a | 0.05 a | 0.00 a | 0.00 a | 0.00 a |
| *Combretum micranthum* | 217 | 1.69 a | 0.10 a | 20.88 b | 1.10 a | 7.50 a |
| *Daniellia oliveri* | 34 | 0.23 a | 1.48 a | 0.00 a | 0.00 a | 0.00 a |
| *Detarium microcarpum* | 3 | 0.23 a | 0.00 a | 0.00 a | 0.00 a | 0.00 a |
| *Diospyros mespiliformis* | 172 | 0.00 a | 2.14 a | 0.88 a | 1.00 a | 55.00 b |
| *Eucalyptus camaldulensis* | 182 | 0.00 a | 8.48 a | 0.00 a | 0.40 a | 0.00 a |
| *Feretia apodanthera* | 26 | 0.31 a | 0.05 a | 1.88 b | 0.60 a | 0.00 a |
| *Ficus* sp. | 1 | 0.00 a | 0.05 a | 0.00 a | 0.00 a | 0.00 a |
| *Guiera senegalensis* | 3426 | 17.00 a | 25.10 a | 313.63 b | 6.10 a | 54.00 a |

| Taxon | Total Abundance of Woody Plants | Petric Pisolithic Plinthosols N = 13 | Petric Plinthosol N = 21 | Lixisol N = 8 | Pisolithic Plinthosol N = 10 | Gleysols N = 2 |
|---|---|---|---|---|---|---|
| *Khaya senegalensis* | 2 | 0.00 a | 0.10 a | 0.00 a | 0.00 a | 0.00 a |
| *Lannea microcarpa* | 108 | 1.38 a | 1.62 a | 1.25 a | 3.10 a | 7.50 a |
| *Maytenus senegalensis* | 1 | 0.00 a | 0.00 a | 0.00 a | 0.00 a | 0.50 b |
| *Parkia biglobosa* | 14 | 0.31 a | 0.19 a | 0.00 a | 0.20 a | 2.00 b |
| *Piliostigma reticulatum* | 439 | 3.69 a | 3.43 a | 26.13 b | 5.70 a | 26.50 a |
| *Saba senegalensis* | 2 | 0.00 a | 0.05 a | 0.13 a | 0.00 a | 0.00 a |
| *Sclerocarya birrea* | 5 | 0.00 a | 0.14 a | 0.13 a | 0.10 a | 0.00 a |
| *Senegalia pennata* | 46 | 1.46 a | 0.00 a | 3.38 b | 0.00 a | 0.00 a |
| *Sterculia* sp. | 2 | 0.08 a | 0.05 a | 0.00 a | 0.00 a | 0.00 a |
| *Tamarindus indica* | 1 | 0.00 a | 0.00 a | 0.13 a | 0.00 a | 0.00 a |
| *Terminalia* sp. | 89 | 0.00 a | 0.00 a | 10.75 b | 0.00 a | 1.50 a |
| *Vachellia seyal* | 24 | 0.00 a | 0.00 a | 0.00 a | 0.00 a | 12.00 b |
| *Vitellaria paradoxa* | 47 | 0.46 a | 1.48 a | 0.75 a | 0.30 a | 0.50 a |
| *Waltheria indica* | 1 | 0.08 a | 0.00 a | 0.00 a | 0.00 a | 0.00 a |
| *Ximenia americana* | 53 | 0.15 a | 0.00 a | 6.25 b | 0.10 a | 0.00 a |
| Total 26 spp. | 4999 | 27.77 a | 45.57 a | 390.50 b | 20.70 a | 175.00 a |

Within lines, values followed by different letters are significantly different using the Tukey–Kramer test ($\alpha = 0.05$). "N" in the line below the names of the soil types indicates number of quadrats in which the woody species appeared.

### 3.4. Preferred Useful Trees

The ages of the 30 respondents in the rural survey ranged from 39 to 85 years old (median 65 years old). In total, 31 tree species were specified as useful trees by the respondents (however, these are preferred trees and do not exactly match the trees identified in the line transect area). Figure 5 summarizes the answers to Question 1 regarding the relationship between preference for tree uses and tree versatility (VER) and shows the 10 top-ranked tree species. More than 90% of the respondents pointed out that *P. biglobosa*, *V. paradoxa*, and *L. microcarpa* were useful and multipurpose trees. For example, these species are particularly useful as sources of fiber for making rope, edible fruits, traditional medicines, fuels, and sources of income generation. These three species are well known as native fruit trees that are highly marketable. Regarding the parts used, 70–80% of respondents answered that these three species, which cannot be cut down because of the customary law of the village, are also a good fuel source. Residents stated that branches, bark, and leaves were removed (while keeping the trees alive) and partially used as fuel. In addition, 20 of the 30 respondents produced compost from the litter (leaves and branches) of these three species, and 16 respondents used litter from the trees for termite nest farming systems, which were expected to decompose the litter and return the nutrients to the soil. Eleven respondents answered that they incinerate the litter and apply the residue as fertilizer before crops are sown. However, in recent years, the distribution of termite mounds has decreased; therefore, the application of litter to termite mounds is now limited.

With regard to rules and legislation regulating tree use and the right to use trees (Question 2), 28 of the 30 respondents were aware that use of trees is governed by laws and regulations or local customs. Twenty-eight respondents answered that the traditional practices existed before the use of forests was restricted under the National Forest Law, for example, and that the use of trees is regulated in this area at present. Of the aforementioned 28 respondents, 26 respondents recognized that the felling of standing trees was prohibited, but that pruning of branches from trees and the use of leaves and fruits were permitted, whereas the remaining two respondents considered that cutting of standing trees was also allowed.

With respect to conflict among residents on the right of use of trees in the farmed parkland (Question 3), eight of the 30 respondents answered that there was conflict. The conflict was attributed to land ownership (or right of use) issues among residents of the same or different communities.

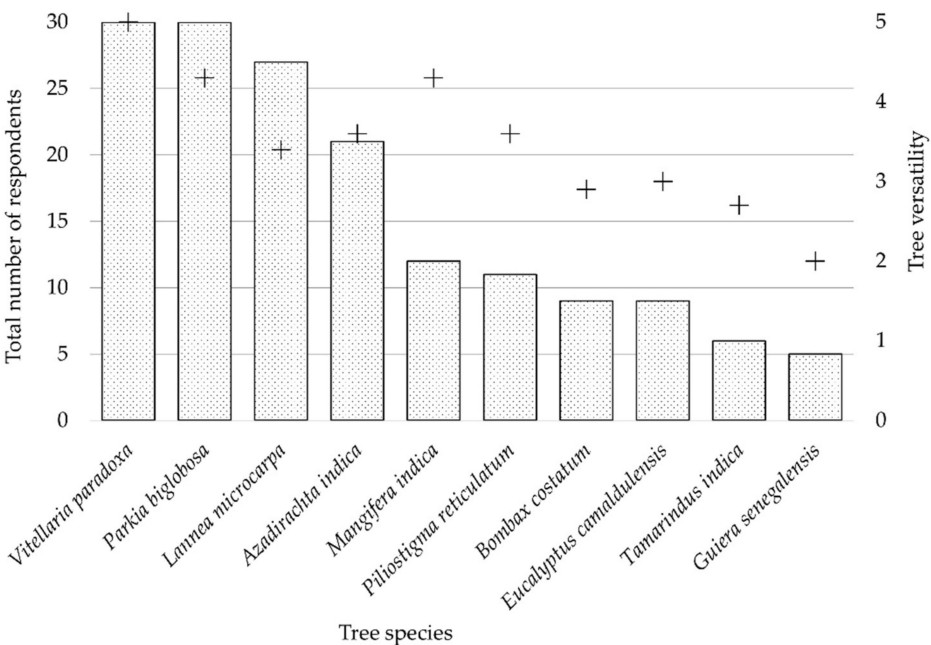

**Figure 5.** The 10 top-ranked species of trees preferred by survey respondents and the versatility of each species. Bars indicate the number of respondents, "+" indicates the tree versatility.

## 4. Discussion

### 4.1. Composition of Wood Vegetation

The overall vegetation composition in the study area was similar to that observed by Weigel [29], which was characteristic of the vegetation in the transition between the Sahelian and Sudanian climate zones. Of the 26 tree species recorded in the current study, 18 species (69%) were also previously reported [29].

The mean total BA of woody vegetation in cultivated land, fallow land, and reserved forest in the south-central region of Burkina Faso was 1.1, 3.3, and 4.3 m$^2$ ha$^{-1}$, respectively [30]. The total BA of trees of DBH $\geq$ 5 cm in the current study (1.6 m$^2$ ha$^{-1}$) is similar to that recorded in cultivated land [30], which indicated that the farmed parkland in the present study site had a similar light environment to cultivated fields with sparse woody vegetation. This finding may be because farmers prefer to keep total CC low to secure light for crops.

The overall $H'$ and $E$ in the present study area were 1.36 and 0.41, respectively (Table 1). These values were relatively low compared with those of previous studies. The $H'$ and $E$ of two parklands in other areas of Burkina Faso were 1.95 and 0.86, and 1.53 and 0.72, respectively [30], which suggests that $H'$ in farmed parklands of Burkina Faso is limited to 1−2 with higher $E$ values, and with considerably greater equality than those of the present study site. The $H'$ ($E$) for three degraded forests, 10 primary forests, and one agroforestry site in Northeastern Congo are 3.54−4.33 (0.79−0.93), 2.46−4.47 (0.76−0.89), and 4.12 (0.95), respectively [31]. The present $H'$ ($E$) values differed substantially from those of all Congolese forest types, including from the agroforestry site, which must be often used by the residents. This difference may be because residents in Burkina Faso farm the land much more constantly and particularly utilize useful trees [32]. Consequently, useful tree species are frequently selected, which reduces diversity and evenness caused not only by their usefulness and local customary law but also national law stipulations.

The three dominant tree species determined by IVI (*V. paradoxa*, *L. microcarpa*, and *P. biglobosa*; Table 2) are well known as useful trees that provide fruit [33], medicines, and fodder [34]. Therefore, the dominance of these trees is likely closely associated with human needs and activities. Trees with a high IVI were seldom harvested except for the exploitation of small branches.

### 4.2. Relationships between Topographic Position, Soil Type, and Woody Vegetation

The mean number of individuals of tree species was lower on Plinthosols and higher on Lixisols (Table 3). This can be explained by the differences in soil water condition between these soil types. Plinthosols generally have poor soil water conditions because the effective soil depth is limited by one or both of the petroplinthic and pisoplinthic horizons, whereas Lixisols in the lower portion of the transect show better soil water conditions because the effective soil depth is greater and the water table is higher. On Gleysols in the valley bottom, tree growth appears to be limited by waterlogging, so the mean number of plants was not higher than that on Lixisols, although Gleysols have a higher water table than Lixisols. The soil toposequence was similar to that reported previously [15].

At least two of *G. senegalensis*, *P. reticulatum*, and *C. micranthum* were dominant in all soil types and topographic positions. These species are indicators of poor soils in West Africa [27,29], indicating that the farmed parkland in the current study has degraded soils. Severe water erosion has previously been reported on the Central Plateau of Burkina Faso [35,36]. However, as recommended by Lahmar et al. [37], regenerated *G. senegalensis* and *P. reticulatum* have the potential to assist in the rehabilitation of degraded land and improve primary productivity.

*Vitellaria paradoxa* and *L. microcarpa*, two of the dominant tree species based on IVI (Table 2), were distributed widely along the transect and did not vary significantly in abundance among the five soil types. The results suggest that residents have selected and conserved these useful trees. Other human influences may have affected the composition of the woody vegetation. *Vitellaria paradoxa* is widely distributed on any type of soil from Sudanian to Guinean savannas, and *L. microcarpa* is distributed on a variety of soil types from Sahelo-Sudanian to Sudanian savannas [27]. Tree species composition has changed substantially towards increased dominance of drought-resistant species after the severe droughts in the 1970s and 1980s in the farmed parklands of Northern Burkina Faso, where there is less precipitation than in the southern zone (the site of the present study). This change in tree species composition in the parkland represents a shift towards more drought-tolerant woody vegetation during a period of increasing annual rainfall. In addition, intensive land management can lead to greater tree cover closer to houses [17]. Limited information is available to explain the mechanism of drought tolerance of *V. paradoxa* and *L. microcarpa*. However, one possible reason is the deciduousness of the two species, which differ in phenological patterns. *Vitellaria paradoxa* is a deciduous or semi-deciduous tree with a defoliation period of 0–2 months during March and April, whereas *L. microcarpa* is a deciduous tree with a full defoliation period of 3–5 months during November–March. Deciduousness is an outcome of the integrated effect of drought, tree characteristics, and soil moisture conditions [38]. Thus, deciduousness can be a reliable indicator of seasonal drought experienced by different tree species. The reason why these two tree species are widely distributed in the current study site on different soil types may reflect that deciduousness is a favorable adaptation to the integrated natural conditions in conjunction with the intervention of intensive tree use by local residents. On Lixisols along the lower transect, we observed fewer useful trees and a higher number of shrubs, e.g., *G. senegalensis*. It is possible that some of this area is under territorial dispute between villages, as such situations can result in the prohibition of tree planting, farming, and browsing, thus encouraging shrub regeneration.

### 4.3. Changes in Woody Vegetation over the Last 31 Years

The dominant species in the valley bottom and lower transect observed by Guinko (as reported in [20] and summarized by [21]) were the same five species reported herein, all of which are used for multiple purposes (i.e., self-consumption, sale of fruit, and traditional medicines). Three of these species (*Anogeissus leiocarpus*, *Sclerocarya birrea*, and *V. paradoxa*) have declined in abundance, which could be indicative of over-utilization by residents [7]. *Vitellaria paradoxa* has undergone a particularly marked decline from abundant to sparse. Deforestation in this region has been caused by farmland expansion triggered by population

growth and human migration from northern to southern regions of Burkina Faso [39]. In this process, farmed parklands might have experienced degeneration resulting from over-utilization. According to a previous contribution to a national forest inventory in Burkina Faso [40], *V. paradoxa* was considered to be the most important tree species in Burkina Faso, as it was highly abundant and widely distributed in the country. However, the abundance of the tree differed in the three land-use categories "forest", "other wooded land", and "other land (with tree cover)", based on the classification of FAO [41]. Regenerated smaller trees are less frequent in "other wooded land" and "other land" but more frequent in "forest", and trees in non-forest lands usually remain uncut to serve specific utilization purposes, such as pulp collection [40]. Similarly, no small trees of both *V. paradoxa* and *P. biglobosa* were recorded in the present study site (Table 1), which would be classified as "other land" as defined by FAO [41].

Useful trees of high monetary value are well managed in farming lands, but seedlings are rare owing to multiple factors, including wildfires, weeding, and browsing, which contribute to farming and pasture practices. As a result, only individuals of relatively large DBH remain and in time may lead to population decline [3,42,43]. In the present study, no individuals of *V. paradoxa* and *P. biglobosa* of DBH < 5 cm were observed. Therefore, as the number of existing individuals of DBH ≥ 5 cm decreases, the persistence of populations of useful trees will decline in the future. It is recommended that manipulation of the population structure to have a balanced tree-size composition using natural and artificial regeneration methods is considered. Among useful trees, only the *L. microcarpa* population included 72 small (DBH < 5 cm) individuals. It is unclear why the *L. microcarpa* population included more small trees than larger individuals (DBH ≥ 5 cm). It may be that *L. microcarpa* is unpalatable to livestock or the germination requirements for *L. microcarpa* differ from those of *V. paradoxa* and *P. biglobosa*.

*Guiera senegalensis* and *P. reticulatum* showed greater abundance in the lower transect and valley bottom, and *C. micranthum* abundance was higher from the middle transect to the lower and upper transect. It has previously been suggested that these three species are indicators of poor soil and that *G. senegalensis* and *C. micranthum* can be dominant on degraded or poor soils in West Africa [27,29]. Moreover, integrating those two shrubs in a farming system can assist in soil rehabilitation and improve primary productivity [37]. Therefore, we concluded that soil erosion and degradation have affected the composition of woody vegetation in the study area during the last 31 years, and the changes in woody vegetation may provide an opportunity to improve degraded lands.

Two exotic tree species rarely encountered in the past (*Azadirachta indica* and *Eucalyptus camaldulensis*) have become dominant in certain soil types and topographic locations. Furthermore, *Adansonia digitata* (baobab) was not observed in the current study. According to Boffa [3], *Azadirachta indica* was introduced to Sahelian countries in the late 1910s because of its high tolerance on lateritic and shallow soils under low rainfall and because of its multiple uses. *A. indica* thrives in some parklands such as those in Bulkiemdé Province, Burkina Faso [44]. Farmers first planted this tree on land that was unsuitable for crop production, and then on areas close to compounds [44]. Saplings of *Eucalyptus camaldulensis* were widely provided for planting after 1973, and it turned out to be the best performing species in terms of hardiness (adaptation to poor soils and aridity) and fast growth [45,46]. We presume that these two species were not planted among scattered useful native trees in the farmed parkland, but mainly in the degraded areas and compounds of the residents. These changes in the dominant species in the farmed parkland may be due to changes in the local inhabitants' dependence on or preference towards natural resources over time.

## 5. Conclusions

The composition of the woody vegetation in a farmed parkland of central Burkina Faso has greatly changed from 1984 to 2015. This change is likely to be caused by soil erosion, land degradation, and changes in the local residents' dependence on or preference towards the woody vegetation. Native fruit trees, such as the shea-butter tree (*V. paradoxa*),

which are highly commodifiable and versatile, are important natural resources beneficial for rural residents and vital for the ecosystem services and functions of farmed parkland in the study area. The expansion of ecological indicator plants in degraded areas, such as *G. senegalensis*, suggests that soil degradation has intensified in recent decades and has impacted the vegetation composition. A survey of 30 households revealed the current status and background information on the preferences and uses of trees by local residents in the study area. The absence of similar information from the area from the preceding 31 years precludes interpretation of whether preferences and uses of trees have changed. The interviews revealed that the residents understood that forest conservation regulations exist and that the use of trees is regulated. Tree use is expected to be managed appropriately if not excessively, but the most preferred tree species, which are highly salable, are extremely slow to mature. The farmed parkland is experiencing a regeneration crisis because sapling survival is threatened by multiple factors, such as wildfires and browsing. To permanently conserve these resources, which play a vital role in rural livelihoods, it is essential to improve and disseminate regreening technology as well as raise awareness of the importance of sustainable land and tree use among residents.

**Author Contributions:** Conceptualization, K.T. and K.I.; methodology, K.T. and K.I.; software, K.T. and K.I.; validation, K.T. and K.I.; formal analysis, K.T. and K.I.; investigation, K.T., K.I., S.S., F.K., N.T., and J.K.; resources, S.S., F.K., N.T., and J.K.; data curation, K.T. and K.I.; writing—original draft preparation, K.T. and K.I.; writing—review and editing, K.T. and K.I.; visualization, K.T. and K.I.; supervision, K.T.; project administration, K.T.; funding acquisition, K.T. and K.I. All authors have read and agreed to the published version of the manuscript.

**Funding:** This study received no external funding.

**Data Availability Statement:** All data analyzed are included in this article.

**Acknowledgments:** This study presents results from the research project "Development of watershed management model in the Central Plateau, Burkina Faso" implemented by the Japan International Research Center for Agricultural Sciences. We thank Robert McKenzie, from Edanz Group (https:// en-author-services.edanz.com/ac (accessed on 31 March 2021)), for editing a draft of this manuscript.

**Conflicts of Interest:** The authors declare no conflict of interest.

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
