# Peer review of "Changes in Woody Vegetation over 31 Years in Farmed Parkland of the Central Plateau, Burkina Faso"

_land, doi:10.3390/land10050470_

Round 1
Reviewer 1 Report
Dear authors,
Thank you for addressing most of the issues raised. I have made some minor comments, which I hope you will find useful and constructive.

Reviewer 2 Report
I reviewed this paper and find this paper is easy to read and understand. Pleaded manuscript seems to be a draft . Comments are visible, inserted and corrected text, font format and color is different. The font in the figures and tables should also be changed, now it is illegible. Figure captions should not be transferred to the next page. In terms of editing, the article should be worked on and adapted to the requirements of the editorial office.
My primary concern, is the lack of utility of the survey presented in this study. Carrying out a survey on a small group of respondents (30 people) does not give results on the basis of which real conclusions can be drawn. I suggest to remove the part relating to the survey results from the manuscript.
Comparison with previous research (previous information)- This seemingly belongs in the discussion section, no materials and methods
Author Response
Please see the attachment.

This manuscript is a resubmission of an earlier submission. The following is a list of the peer review reports and author responses from that submission.
Round 1
Reviewer 1 Report
The manuscript “Changes in Woody Vegetation over 31 Years in Farmed Parkland of the Central Plateau, Burkina Faso” analyzed changes in vegetation composition in agroforestry parklands since 1984 and assessed local community tree uses and preferences in the Central Plateau of Burkina Faso. It characterized woody vegetation abundance along a transect with differential biophysical features. The manuscript is well written, technically sound and will be very relevant to the readership of the journal and I would recommend its publication after issues highlighted below are adequately addressed.
Introduction:
Line 32-43: Parklands represent an agroforestry system widespread in semi-arid areas of West Africa. Boffa and other have provided a clear definition of this system and its socioeconomic and ecological benefits. The discussion about the definition is irrelevant. Please go straight to the point, i.e., changes in tree composition in parklands as driven by human activities.
Boffa, J. M. (1999). Agroforestry parklands in sub-Saharan Africa. Rome, Italy: Food and Agriculture Organization of the United Nations (FAO).
Line 51-67: Climate change has also played a role in change in vegetation composition change in the region including in Burkina Faso. Please update this section to reflect the current state of knowledge
Hänke, H., Börjeson, L., Hylander, K., & Enfors-Kautsky, E. (2016). Drought tolerant species dominate as rainfall and tree cover returns in the West African Sahel. Land Use Policy, 59(Supplement C), 111-120. doi: 10.1016/j.landusepol.2016.08.023
Zida, W. A., Traoré, F., Bationo, B. A., & Waaub, J.-P. (2020). Dynamics of woody plant cover in the Sahelian agroecosystems of the northern region of Burkina Faso since the 1970s–1980s droughts. Canadian Journal of Forest Research, 50(7), 659-669. doi: 10.1139/cjfr-2019-0247
Zida, W. A., Bationo, B. A., & Waaub, J.-P. (2020). Re-greening of agrosystems in the Burkina Faso Sahel: greater drought resilience but falling woody plant diversity. Environmental Conservation, 47(3), 174-181. doi: 10.1017/S037689292000017X
Brandt, M., Hiernaux, P., Rasmussen, K., Tucker, C. J., Wigneron, J.-P., Diouf, A. A., . . . Fensholt, R. (2019). Changes in rainfall distribution promote woody foliage production in the Sahel. Communications Biology, 2(1), 133. doi: 10.1038/s42003-019-0383-9
Brandt, M., Rasmussen, K., Hiernaux, P., Herrmann, S., Tucker, C. J., Tong, X., . . . Fensholt, R. (2018). Reduction of tree cover in West African woodlands and promotion in semi-arid farmlands. Nature Geoscience, 11(5), 328-333. doi: 10.1038/s41561-018-0092-x
Brandt, M., Rasmussen, K., Peñuelas, J., Tian, F., Schurgers, G., Verger, A., . . . Fensholt, R. (2017). Human population growth offsets climate-driven increase in woody vegetation in sub-Saharan Africa. Nature Ecology & Evolution, 1, 0081. doi: 10.1038/s41559-017-0081
Brandt, M., Tappan, G., Diouf, A. A., Beye, G., Mbow, C., & Fensholt, R. (2017). Woody Vegetation Die off and Regeneration in Response to Rainfall Variability in the West African Sahel. Remote Sensing, 9(1). doi: 10.3390/rs9010039
Spiekermann, R., Brandt, M., & Samimi, C. (2015). Woody vegetation and land cover changes in the Sahel of Mali (1967–2011). International Journal of Applied Earth Observation and Geoinformation, 34(Supplement C), 113-121. doi: 10.1016/j.jag.2014.08.007
Zhang, W., Brandt, M., Tong, X., Tian, Q., & Fensholt, R. (2018). Impacts of the seasonal distribution of rainfall on vegetation productivity across the Sahel. Biogeosciences, 15(1), 319-330. doi: 10.5194/bg-15-319-2018
West, C. T., Moody, A., Nébié, E. K., & Sanon, O. (2017). Ground-Truthing Sahelian Greening: Ethnographic and Spatial Evidence from Burkina Faso. Human Ecology, 45(1), 89-101. doi: 10.1007/s10745-016-9888-8
Line 68-69: This reflects at best a lack of knowledge of literature in the region. There are many scientific papers out there on woody vegetation composition and changes in the region including in Burkina Faso.
Methods
Line 164: Why did you limit the probing to 10 species? Or is it 10 species that were recorded during tree inventory??? A total of 26 species was mentioned in the result section
Line 166-167: It would be good if you indicate how you come up with these use groups
Line 168-171: Please provide a reference to support the computation of the versatility index. Keep in mind that there are several ethnobotanical indices that are relevant to assess the most frequent tree species.
Coe, M. A., & Gaoue, O. G. (2020). Most Cultural Importance Indices Do Not Predict Species’ Cultural Keystone Status. Human Ecology, 48(6), 721-732. doi: 10.1007/s10745-020-00192-y
Gaoue, O. G., Coe, M. A., Bond, M., Hart, G., Seyler, B. C., & McMillen, H. (2017). Theories and Major Hypotheses in Ethnobotany. Economic Botany, 71(3), 269-287. doi: 10.1007/s12231-017-9389-8
Line 172-175: This is a repetition of the third sentence of this paragraph. Please edit out one
Results
Line 199-205: This sentence is too long. Please break it down into two or three.
Line 205-206: This is not relevant here. Please move it to Discussion section.
Table 2: Why did you limit to only 11 species? I recommend to provide growth parameter for all the 26 species identified
Line 275 & 278: Please include those data. There is no page limitation for this journal. It can also be shown in supplementary materials
Line 288-290: I am a bit confused: in the method section, it was mentioned that respondents were asked about uses of 10species while 26 species were identified from the inventory.
Line 335: What does this stand for? Please define at first use.
Line 351-355: Please consult the references below for strengthening this section.
Gaoue, O. G., Coe, M. A., Bond, M., Hart, G., Seyler, B. C., & McMillen, H. (2017). Theories and Major Hypotheses in Ethnobotany. Economic Botany, 71(3), 269-287. doi: 10.1007/s12231-017-9389-8
Segnon, A. C., Achigan-Dako, E., Gaoue, O., & Ahanchédé, A. (2015). Farmer’s Knowledge and Perception of Diversified Farming Systems in Sub-Humid and Semi-Arid Areas in Benin. Sustainability, 7(6), 6573-6592. doi: 10.3390/su7066573
Line 366-370: The first two species are also used for soil restoration. This traditional knowledge based technique is described by Lahmar et al 2012 and offers avenues for tailored agricultural intensification. Please read the paper below
Lahmar, R., Bationo, B. A., Dan Lamso, N., Guéro, Y., & Tittonell, P. (2012). Tailoring conservation agriculture technologies to West Africa semi-arid zones: Building on traditional local practices for soil restoration. Field Crops Research, 132(0), 158-167. doi: 10.1016/j.fcr.2011.09.013
Line 377-380: Please update this section to reflect the state of knowledge.
Hänke, H., Börjeson, L., Hylander, K., & Enfors-Kautsky, E. (2016). Drought tolerant species dominate as rainfall and tree cover returns in the West African Sahel. Land Use Policy, 59(Supplement C), 111-120. doi: 10.1016/j.landusepol.2016.08.023
Line 382: Please discuss more this driver. there are several literature pointing towards the role of climate change in vegetation composition change in the region. I mention a few
Line 434-437: Please be nuanced here. There are evidence of traditional soil restoration based on these two species. Please read Lahmar et al 2012.
Lahmar, R., Bationo, B. A., Dan Lamso, N., Guéro, Y., & Tittonell, P. (2012). Tailoring conservation agriculture technologies to West Africa semi-arid zones: Building on traditional local practices for soil restoration. Field Crops Research, 132(0), 158-167. doi: 10.1016/j.fcr.2011.09.013
I have provided some edits directly in the annotated PDF file enclosed

Reviewer 2 Report
Very interesting paper that increases the knowledge on central African agroforestry systems. But the authors need to pay attention to the following points:
-comparisons using sentences such as "data not shown" lines 274-278-279 cannot be accepted in a scientific journal, so the text need to be re-phrased.
-some typing mistakes table 1 pg 7 is overwritten, table 2 need to be re-alligned
Reviewer 3 Report
Dear authors,
Please see the attachment, where I have made several comments and suggestions which I hope you will find useful and constructive
